# Unwinding BRAHMA Functions in Plants

**DOI:** 10.3390/genes11010090

**Published:** 2020-01-13

**Authors:** Caroline Thouly, Marie Le Masson, Xuelei Lai, Cristel C. Carles, Gilles Vachon

**Affiliations:** Université Grenoble Alpes, unité de formation et de recherche de Chimie et Biologie, Institut national de recherche pour l’agriculture, l’alimentation et l’environnement (INRAe), Centre National de la Recherche Scientifique (CNRS), Commissariat à l’énergie atomique et aux énergies alternatives (CEA), Institut de recherche interdisciplinaire de Grenoble (IRIG), Laboratoire de Physiologie Cellulaire et Végétale, 38000 Grenoble, France; caroline.thouly@cea.fr (C.T.); marie.lemasson@cea.fr (M.L.M.); xuelei.lai@cea.fr (X.L.); christel.carles@univ-grenoble-alpes.fr (C.C.C.)

**Keywords:** BRAHMA, SWI2/SNF2, chromatin, transcription factor, remodeling, miRNAs

## Abstract

The ATP-dependent Switch/Sucrose non-fermenting (SWI/SNF) chromatin remodeling complex (CRC) regulates the transcription of many genes by destabilizing interactions between DNA and histones. In plants, BRAHMA (BRM), one of the two catalytic ATPase subunits of the complex, is the closest homolog of the yeast and animal SWI2/SNF2 ATPases. We summarize here the advances describing the roles of BRM in plant development as well as its recently reported chromatin-independent role in pri-miRNA processing in vitro and in vivo. We also enlighten the roles of plant-specific partners that physically interact with BRM. Three main types of partners can be distinguished: (i) DNA-binding proteins such as transcription factors which mostly cooperate with BRM in developmental processes, (ii) enzymes such as kinases or proteasome-related proteins that use BRM as substrate and are often involved in response to abiotic stress, and (iii) an RNA-binding protein which is involved with BRM in chromatin-independent pri-miRNA processing. This overview contributes to the understanding of the central position occupied by BRM within regulatory networks controlling fundamental biological processes in plants.

## 1. Introduction

The DNA of eukaryotic genomes is packaged into a highly organized nucleoprotein complex called chromatin. The first order of chromatin compaction consists of nucleosomes made up of 146/147 bp of DNA wrapped around histone octamers with two copies of each histones H2A, H2B, H3, and H4 together with intervening histone-free linker DNA [1]. This packaging generates a barrier and needs to be timely and spatially relieved for processes that require access to the DNA, such as transcription. One of the main strategies by which cells alleviate chromatin-mediated repression is through the action of ATP-dependent chromatin remodeling complexes (CRCs) that destabilize interactions between DNA and histones. The most exhaustively studied class of ATP-dependent CRCs is Switch/Sucrose non-fermenting (SWI/SNF) and its catalytic ATPase subunit Swi2p/Snf2p that was originally found in yeast and was later identified as BRAHMA (BRM) in Drosophila [2].

The main remodeling activities of the SWI/SNF complex involves both sliding of nucleosomes in addition to the disruption of histone–DNA interactions [3]. This activity can be performed in vitro by the core complex, formed by the SWI2/SNF2-type ATPase, one SNF5, and two copies of SANT/SWIRM/Leu zipper—containing proteins termed SWI3 subunits [4].

The Arabidopsis genome encodes four putative SWI/SNF-type SNF2-ATPases (BRM/CHROMATIN REMODELING 2 (CHR2), SPLAYED (SYD/CHR3), MINUSCULE 1 (MINU1/CHR12) and MINUSCULE 2 (MINU2/CHR23)), a single SNF5 subunit called BUSHY (BSH), four SWI3 subunits (SWI3A–D), two BRM-associated factors (SWP73A and B/BAF60), two actin-related proteins (ARP4 and ARP7) and some plant-specific subunits such as the co-repressor LEUNIG [5].

Analysis of the two closely related ATPases, BRM and SYD, showed that they have both unique and redundant functions in plants [6]. In contrast with their metazoan orthologs, null mutants of BRM and SYD are viable but fertile, facilitating investigation of their roles throughout plant development. Mutants of either *BRM* or *SYD* leads to many developmental defects, including slow growth rate, defects in cotyledon separation, and reduced apical dominance and flower morphogenetic defects [7,8,9,10]. Null mutants in *BRM* alone have unique root growth defects [7,9,10]. Double mutants of *BRM* and *SYD* cause embryo lethality mainly at the heart stage [6].

Here, we review genetic and biochemical data that have shed new light on the function of BRM in a diverse array of developmental processes and emphasize the roles of non-SWI/SNF protein partners of BRM involved in each of these processes. This allows us to distinguish three types of BRM interactors, that play a role either in addressing BRM to specific genomic loci, in regulating BRM stability or in chromatin-independent pri-miRNAs processing. The latter role opens new exciting and unique opportunities to understand the role of BRM in all eukaryotes.

## 2. BRM Functional Domains Are Conserved in Plants

BRM is a large protein of 2193 residues and has the canonical domains found in this family of proteins (Figure 1a) [11]. BRM contains an N-terminal region with a glutamine-rich region, a glutamine-leucine-glutamine (QLQ)-rich region and a helicase SANT-associated domain (HSA) as well as an adjacent domain termed A-HSA [12]. While the function of Q and QLQ-rich regions remains unclear, the HSA domain in human BRM (BRG1) seems to mediate the interaction with the BAF250a/ARID1A subunit of human SWI/SNF complex and plays a role in transcriptional activation of downstream genes [13]. Downstream of the HSA domain, BRM displays the central catalytic helicase-like ATPase domain composed of the DEXHc ATP-binding domain and the HELICc domain, both found in a diverse family of proteins helicases. The C-terminal region of BRM contains an AT-hook that allows BRM to bind DNA [14] and a bromodomain which typically recognizes acetylated lysine residues, such as those on the N-terminal tails of histones [15]. The latter is absent in the three other ATPases of Arabidopsis making BRM the closest homolog of the yeast and animal ATPases. A detailed biochemical analysis of the C-terminal region of BRM identified several double-strand DNA binding and nucleosome binding regions, in addition to the AT-hook region (Figure 1a) [14]. The *brm-3* allele which lacks two of the three identified domains, as well as the bromodomain, has a moderate phenotype, indicating that these domains are required for normal BRM functions. Although the full length BRM protein has not been crystallized, structures of the central region [16] and bromodomain [17] have been described in humans.

Although no SWI/SNF chromatin-remodeling complex has been purified to homogeneity, several studies support the interaction of BRM with members of CRC subunits in Arabidopsis. SWI3C seems to be a dedicated BRM partner and complex component, based on their physical interaction and the highly similar loss-of-function phenotypes of *swi3c* and *brm* mutants [11,18]. SWI3A and SWI3B could also participate to the complex through direct interaction with BRM [11]. Other subunits such as BSH, ARPs, or SWP73s could also be subunits of the BRM complex through interaction with SWI3 proteins.

## 3. Genome-Wide Functions of BRM in Plants

*BRM* gene is expressed primarily in meristems and proliferating tissues such as inflorescences, calli and cultured cells and, to a lesser extent, in leaves and can be detected in siliques, root, and seedlings [8]. In Drosophila, the unique *BRM* gene has been classified as a member of the *trithorax* group (trxG) for its ability to suppress the phenotype of mutations in *Polycomb* group (PcG) genes through regulation of homeotic genes [2,19,20]. In Arabidopsis, BRM is also required for proper expression of homeotic genes in flowers [9] and can balance the role of PcG genes in vegetative development [21]. The PcG Polycomb Repressive Complex 2 (PRC2) complex catalyzes the tri-methylation of histone 3 (H3) at lysine 27 (H3K27me3), the hallmark of gene repression in eukaryotes. Genome-wide H3K27me3 profiling in seedlings showed that BRM reduces H3K27me3 level at several hundred genes, a role that is partially suppressed by removal of the H3K27 methyltransferase CURLY LEAF (CLF) or SWINGER (SWN) [22], which are the catalytic subunits of PRC2. Together, these results indicate a crucial role of BRM in restricting the repressive activity of PcG during plant development [22].

BRM was shown to be recruited to specific loci by direct physical interaction with histone demethylase RELATIVE OF EARLY FLOWERING 6 (REF6) [23], a plant-unique H3K27me3 demethylase that antagonizes Polycomb group activity at target loci [24] and contains several zinc-finger motifs that targets genomic loci containing a CTCTGYTY motif. Indeed, genome-wide analyses showed that REF6 colocalizes with BRM at many genomic sites with the CTCTGYTY motif and facilitates the recruitment of BRM. This strongly suggests a mechanism for SWI2/SNF2 ATPase recruitment to these loci in which a specific DNA-binding protein precedes the remodeler at the target loci.

In addition to its site-specific recruitment, BRM has recently been shown to have an overlapping role with the H2A variant H2A.Z with both antagonistic and cooperative contributions, especially on environmentally responsive genes [25]. The authors identified eight classes of genes that show distinct relationships between H2A.Z and BRM with respect to their roles in transcription. These include redundant and antagonistic interactions between BRM and H2A.Z. 

Using ChIP-on-chip experiments combined with transcriptional profiling of the *brm-1* mutant, Archacki et al. [26] showed that BRM activates and represses nearly as many target genes. Interestingly, while BRM was enriched near transcriptional start sites (TSS) of target genes as expected, many target genes were bound at their 3′ end, especially at 3′ regions that have promoter-like structures such as TATA boxes, high H3K4me3 levels, low H3K27me3 and di-methylated lysine 9 on H3 (H3K9me2) levels, and high antisense transcriptional activity. However, BRM does not co-localize with the H3K27me3 mark [22] but with H3K9 and H3K14 acetylation marks [27].

## 4. BRM Is Required throughout Plant Development

### 4.1. BRM Is Involved in Vegetative Development

The activity of BRM and SYD is necessary for proper embryo development as *brm syd* double-mutants show embryo lethality mainly at the heart stage [6]. BRM seems to play a major role in this process since the single *brm-101* mutant shows about 2% of embryo lethality while *syd* embryo lethality is close to wild-type.

In dicotyledenous plants, the cotyledon boundaries and the shoot apical meristem (SAM) are formed at the heart stage. The role of BRM and SYD in this process was identified in a phenotypical enhancer screen using a null allele of the *CUP-SHAPED COTYLEDON 2* (*CUC2)* [10]. It was shown that *BRM* and *SYD* mutations enhance the cotyledon fusion phenotype of *CUC* mutants and *SHOOT MERISTEM-LESS* (*STM*) mutants. BRM upregulates all three *CUC* genes during embryogenesis while SYD only regulates *CUC2*. In addition, REF6 and BRM both regulate the *CUP-SHAPED COTYLEDON 1* (*CUC1*) gene [10,28].

In addition to its role in *CUC* genes regulation, BRM is also involved in several aspects of leaf development. In Arabidopsis, the vegetative phase consists of a juvenile phase followed by an adult phase with the latter being characterized by changes in the production of trichomes on the abaxial side of the leaf blade and by modified leaf characteristics such as an increase in the degree of serration of the leaf margin. The *brm-7* allele was isolated in a forward genetic screen for vegetative phase change modifiers [21]. This mutant exhibits an accelerated phase change by reducing the expression of miR156 which negatively regulates the expression of a class of plant-specific *SQUAMOSA PROMOTER BINDING PROTEIN LIKE* (*SPL*) genes that function as master regulators of vegetative phase. While BRM binds to the *MIR156A* promoter, nucleosome occupancy in the *brm-7* mutant, is increased at −2 and +1 nucleosomes proximal to the TSS of *MIR156A* locus while positioning is not affected. Also, the PRC2 methyl-transferase subunit SWN, but not CLF, functions with BRM to antagonistically regulate the level of H3K27me3 at the *MIR156A* locus [21]. Therefore, BRM plays a role in vegetative phase change in Arabidopsis by directly fine-tuning the expression of *MIR156*. Interestingly, H2A.Z also promotes the expression of *MIR156A/MIR156C* and juvenile phase identity. However, this seems to be achieved by facilitating the deposition of H3K4me3, rather than by decreasing nucleosome occupancy [29].

BRM is involved in the regulation of photomorphogenesis by opposing the effect of the basic helix-loop-helix PHYTOCHROME-INTERACTING FACTOR 1 (PIF1) factor [30]. While *BRM* transcription is down-regulated by light in young seedlings, it negatively regulates greening of etiolated seedlings by direct repression of the *protochlorophyllide oxidoreductase C* (*PORC*) gene. This regulation is PIF1-dependent and likely involves a direct interaction as BRM interacts physically with PIF1 [30]. Similar to *KNAT2* and *KNAT6* regulations, the H3K4me3 level is increased in *brm* plants and the H3K27me3 level is unchanged at *PORC* locus. Interestingly, the BAF60/SWP73B/CHC1 subunit of the SWI/SNF complex is also regulated by light and represses seedling growth by opposing in *cis* the photomorphogenic effect of PIF4 [31]. Later during leaf development, BRM, AtSWI3C, and BSH5 all act as repressors of seed maturation genes (*2S* family) in leaves and bind to their promoters [32].

During leaf development, BRM participates to the regulation of cell proliferation processes regulated by the transcriptional co-activator ANGUSTIFOLIA3 (AN3) [33]. The SWI/SNF complex interacts closely with AN3 as many subunits such as BRM, SYD and BAF60/SWP73B/CHC1 were immunoprecipitated with AN3 from plant material. Furthermore, BAF60/SWP73B/CHC1 is recruited by AN3 to its target genes and *BRM* genetically interacts with *AN3*. BRM is also essential in the expression of AN3 target genes such as *GROWTH REGULATING FACTOR 5 (GRF5)* and *HECATE1* (*HEC1*). Whether AN3 and BRM interact physically has not been reported in the literature. Taken together, this dataset shows that SWI/SNF-AN3 module acts as a major player at the transition from cell proliferation to cell differentiation in a developing leaf.

### 4.2. BRM Controls Flowering Time

The different genetic pathways that control flowering are well defined and chromatin structure plays an important role in such regulations [34].

Arabidopsis plants deficient for BRM activity flower earlier in long days (LD), suggesting that BRM is a repressor of flowering induction. However, analysis of key regulatory flowering genes indicate that BRM has a dual and antagonistic role in the regulation of flowering genes: BRM represses expression of the flowering promoting genes *CONSTANS* (*CO*), *FLOWERING LOCUS T* (*FT*), and *SUPPRESSOR OF OVEREXPRESSION OF CO1* (*SOC1*) which belong to the photoperiod pathway and at the same time of the flowering repressor *FLOWERING LOCUS C* (*FLC*) [35], the main target of the autonomous and vernalization pathways (Figure 1b). In short days (SD), BRM mutants flower late with a proportion of plants that do not flower [35]. The repression of *FLC* by BRM is vernalization-independent suggesting that BRM acts within the autonomous pathway to regulate *FLC* expression [35]. In addition, BRM seems to impose a repressive chromatin configuration at the *FLC* locus as *brm* mutants show increased levels of tri-methylated lysine 4 on histone H3 (H3K4me3) which is a gene activation mark in eukaryotes and a decreased level of H3K27me3.

The regulation by BRM of the key flowering repressor gene *SHORT VEGETATIVE PHASE* (*SVP*) can provide an explanation for the early flowering phenotype of *brm* mutants in LD despite the up-regulation of *FLC* [22]. Indeed, the FLC-SVP complex acts as a repressor of *FT* and the downregulation of *SVP* in *brm* mutants probably prevents *FT* repression, regardless of the increased expression of *FLC*, thus leading to early flowering of *brm* plants.

BRM can also act as a positive regulator of flowering as it is recruited by SOC1, together with the histone demethylase REF6, to oppose PRC2 function at the locus of the flowering inducer *TARGET OF FLC AND SVP1* (*TFS1*) that encodes a B3-type transcription factor [36]. BRM recruitment by SOC1 to the *TFS1* locus is also necessary for the binding of the age-regulated transcription factor SQUAMOSA PROMOTER BINDING PROTEIN-LIKE 9 (SPL9). Nucleosome positioning at a position that encompasses the binding site for SOC1 is destabilized in Col in a REF6- and BRM-dependent manner.

The overlapping roles of BRM and H2A.Z described above is correlated with studies on the regulation of *FLC.* BRM controls the expression of *FLC* by creating a repressive chromatin configuration of the locus mediated by H2A.Z exclusion [35] and H2A.Z occupancy is correlated with high *FLC* transcript levels [37]. Interestingly, H2A.Z also has a direct role on *FLC* regulation in heat-shock conditions [38] and deposition of H2A.Z is positively involved in the regulation of *FLC* and *FLC-like/MAF* genes [39], indicating that the H2A.Z variant confers an activating status to the *FLC* locus.

### 4.3. BRM Shapes Inflorescence and Flower Development

The first reported *brm* mutant in Arabidopsis was obtained upon a screen to isolate enhancers of a mutant of the floral gene *UNUSUAL FLORAL ORGAN* (*UFO*) [40]. The locus was called *FUSED FLORAL ORGANS3* (*FFO3*) as the corresponding mutant allele (*ffo3-1*), a moderate allele of *brm*, showed fused floral organs. Flowers of *brm* loss-of-function mutants have several developmental abnormalities, including homeotic transformations in the second and third floral whorls as well as reduced gametophytic transmission [9].

BRM and SYD are redundantly required for the activation of floral homeotic genes [12]. Both remodelers are recruited to the regulatory regions of *APETALA3* (*AP3*) and *AGAMOUS* (*AG*) during flower development and physically interact via their N-terminal region with the transcriptional activators LEAFY (LFY) and SEPALLATA3 (SEP3). In the case of LFY, the A-HSA/HSA region is involved. Similar to the case of BRM dependency upon REF6, SYD and probably BRM association with the *AP3* and *AG* regulatory regions seems dependent on LFY and SEP3, indicating again that SWI2/SNF2 ATPase recruitment to these loci is preceded by transcription factor binding at their recognition sites [41,42]. In addition, purification of several MADS-domain transcription factor complexes from inflorescences identified BRM, SYD, SWI3A, SWI3B, and BAF60/SWP73B/CHC1 as co-purifying proteins [43].

BRM and SYD also act as trxG factors in flower development as their requirement in floral patterning can be overcome by partial loss of PRC2 activity and induces a decrease in H3K27me3 together with an increase in H3K4me3 at target homeotic genes at least in the case of SYD [12].

BRM plays a role in inflorescence architecture through its interaction with the class-I KNOTTED1-like homeobox (KNOX) transcription factor BREVIPEDICELLUS (BP) [44] which interacts with the N-terminal region of BRM. Loss-of-function *brm* mutants display inflorescence architecture defects resembling those of *bp* mutants, such as clustered inflorescences, horizontally orientated pedicels, short pedicels and internodes. However, it is not known whether BRM colocalizes with KNOX DNA-binding sites [45] or requires BP to bind target genes. Interestingly, the level of H3K4me3 was increased at BP target genes (*KNAT2* and *KNAT6*) in *brm* mutants while no change in H3K27me3 was observed, suggesting that BRM may associate with a histone H3K4 demethylase when acting together with BP.

From these studies describing the roles of BRM in flowering induction, inflorescence and flower development (Figure 1b), BRM seems to act preferentially through interaction with true transcription factors, suggesting that it acts as a specific actor in the processes rather than a broad remodeler.

### 4.4. BRM Has Roles in Abiotic Stress and Hormone Pathways

The involvement of plant and animal SWI/SNF CRCs and their crosstalk with hormone pathways have been reviewed [5]. Several studies with Arabidopsis indicate that BRM plays an important role in the regulation of major hormone signaling pathways, particularly abscisic acid (ABA) [46,47], gibberellin (GA) [48], cytokinin (CK) [49] and auxin [50].

The involvement of BRM in auxin signaling is well observed in roots where BRM regulates stem cell niche maintenance through the auxin distribution pathway and directly targets several *PIN-FORMED* (*PIN*) genes [50]. This activity of BRM seems to be antagonistic to PcG activity in the regulation of *PIN1* and PIN2 as H3K27me3 levels are lower at these loci in *brm* mutants.

BRM integrates plant responses to abiotic stresses, involving interactions with hormone functions that include ABA. Part of the vegetative growth defects of *brm* mutants such as root growth defects seem to be due to depressed ABA responses and *brm* mutants are hypersensitive to ABA. The role of BRM in ABA responses is mediated, in the absence of drought stress stimulus, by the direct repression of the leucine zipper transcription factor *ABA INSENSITIVE5* (*ABI5)* gene together with nucleosome destabilization at the *ABI5* promoter [46]. Since removal of ABI5 activity from *brm* mutants caused partial rescue of their root growth defect, it is likely that additional factors act in parallel with ABA in root elongation. Among these are factors involved in other hormone pathways such as auxin signaling [50] and it would be interesting to see how BRM can play a role in the cross-talk between these pathways.

The ABA pathway core components, type 2C protein phosphatases (PP2Cs) and sucrose non-fermenting 1-related protein kinases (SnRK2s), directly interact with BRM, leading to changes in its phosphorylation status. Indeed, a phospho-mimetic variant of BRM displays ABA hypersensitivity [47].

At the same time, BRM seems to regulate the corresponding *PP2C* genes under abiotic stress conditions. For instance, salt stress converts *PP2C* gene chromatin from a repressed status to an activated status with nucleosome eviction at their promoter and increase levels of H3K4me3 and H3 acetylation [51]. In loss-of-function *brm-3* plants, PP2C expression is more induced under salt stress, indicating that BRM might play a role in mediating this change in chromatin dynamics.

Similarly, BRM may lead to the promotion of genes negatively regulating apical hook formation such as *AXR2*/*IAA7*, potentially through chromatin remodeling [52] as mutations in the ATPase domain of BRM lead to a decreased sensitivity to auxin agonists.

BRM also plays a direct role in plant response to salt stress and ABA in Arabidopsis, by opposing the effects of group VII Ethylene Response Factor AP2-domain transcription factors (ERFVIIs) [53]. ERFVIIs bind a double GGCGGC *cis*-element in the *ABI5* promoter that overlaps with the BRM binding region [46] and ERFVIIs physically interact with the C-terminal region of BRM *in planta*. These data demonstrate a role for BRM-ERFVII interactions in controlling plant growth via opposing functionalities, perhaps in competition for the same *cis*-elements.

In the GA pathway, BRM regulates independently in both GA biosynthetic and GA responses pathways. This seems to be direct at least for the biosynthetic pathway because BRM occupies the promoters of GA biosynthetic genes such as *GA3ox1* [48].

In the regulation of the CK pathway, BRM and basic-helix-loop-helix (bHLH)-related TEOSINTE BRANCHED 1, CYCLOIDEA, PROLIFERATING CELL FACTOR1/2 (TCP) transcription factors modulate leaf responses to CK to promote determinate leaf growth [49]. BRM physically interacts with TCP4 in Y2H and *in planta*. Both BRM and TCP4 bind the promoter of an inhibitor of CK response gene, *ARR16*, and induce its expression. The binding occurs in a region of the *ARR16* promoter that displays TCP binding sites (GTGGTCCA and TGGTCC). Transcriptomic profiling shows that TCP4 and BRM-regulated genes overlap significantly indicating that many other genes might be regulated by BRM and TCPs. TCP4 also interacts physically with SWI3C. The authors also conducted a Y2H interactome study aimed at identifying TFs that can recruit SWI/SNF complexes to the genomic loci they regulate, using a library of 1400 Arabidopsis TFs as prey [49]. 155 potential interactors of BRM (N-terminal domain) were identified belonging to many different TF families. It is not known whether these interactions are relevant *in planta*. Surprisingly, most of the BRM-interacting factors mentioned in this review such as REF6, LFY, SEP3, or BP were not in the reported list of interactors.

Genome stability is critical for the development and survival of all organisms. The stability of the BRM protein has been highlighted by the involvement of the SUMO modification pathway in the regulation of BRM protein level. BRM was identified as a partner of the SUMO ligase METHYL METHANE SULFONATE SENSITIVITY 21 (AtMMS21/HYP2) in a Y2H screen [54]. AtMMS21 is involved in DNA damage response, meiosis, and stress response [55,56,57]. The mutants of *BRM* and *AtMMS21/HYP2* displayed a similar defect in root development and the protein level of BRM is significantly lower in the *mms21-1* mutant than in wild type. Overexpression of BRM in *mms21-1* partially rescued the developmental defect of roots.

BRM stability is also involved in DNA double-strand breaks (DSBs) tolerance in Arabidopsis [27]. High levels of boron induce DSBs. The authors found that boron also induces histone hyperacetylation, probably through inhibition of histone deacetylase (HDAC) activities. BRM binds to acetylated histone marks via its bromodomain to further cause chromatin opening. In order to protect chromatin, proteasome-mediated degradation of BRM limits this chromatin opening.

BRM has been shown to play a major role in the memory of heat-shock (HS) stress in association with the FORGETTER1 (FGT1) factor [58]. FGT1 is the Arabidopsis orthologue of Strawberry notch (Sno) in Drosophila and human. It harbors an ATP-binding DExD/H-like helicase domain, a Helicase C-like domain, and a PHD finger which allows its binding to the N-terminal region of H3, albeit not in a methylation-specific manner, at least in vitro. In contrast to a general role in HS response, both BRM and FGT1 play a specific role in HS memory gene regulation such as *HSA32* and therefore improve HS response after priming with a transient HS. *brm* mutants are deficient in heat shock memory and display a correct induction of HS memory genes such as *HSA32* after priming, but their expression decreases precociously compared to wild type plants. Since *fgt1* mutants display an accelerated recovery of nucleosome occupancy at HS memory loci, the authors propose that the interaction between FGT1 and BRM mediates stress-induced chromatin memory by preventing nucleosome recovery at these loci, hence enabling sustained expression over time. Genome-wide BRM target genes overlap strongly with FGT1 target genes at TSS. Both BRM and FGT1 are pre-associated with memory genes under non-stress conditions and interact physically as shown by bimolecular fluorescence complementation (BiFC) and co-immunoprecipitation [58]. 

## 5. BRM Plays a Chromatin-Independent Role in pri-miRNA Processing

Recently, a major advance has been made in the understanding of BRM biochemical function as well as in its role *in planta* in Arabidopsis. This work demonstrates that BRM has a role in processing non-coding regulatory RNAs by remodeling of pri-miRNAs and impedes miRNA production [59]. This role is distinct from BRM role in transcriptional regulation of miRNAs such as miR156A [21].

Most miRNAs are processed from long pri-miRNAs by Microprocessor and Dicing complexes which includes at least Dicer-like 1 (DCL1) and Hyponastic leaves 1 (HYL1). SERRATE (SE), a zinc-finger protein is also considered to be a core component of plant Microprocessor because *se* mutations cause pri-miRNA accumulation and miRNA loss in vivo. BRM was initially found to be a partner of SE [59]. The authors observed that in the *BRM* mutant *chr2-1*, 114 of the 365 annotated miRNAs in Arabidopsis were up-regulated, of which nearly 60% were also regulated by SE, making BRM a negative regulator of miRNA accumulation for a large subset of miRNAs in Arabidopsis. The authors then underwent a series of in vitro experiments with the reconstituted recombinant Microprocessor (DCL1-HYL1-SE) and showed that BRM inhibits pri-miRNAs processing in a SE-dependent manner. They found that BRM binds directly to pri-miRNAs with a higher affinity than SE, but not higher than HYL1. A BRM-SE-pri-miRNA complex seemed to form in vitro except when HYL1 was present readily sequestering pri-miRNAs from SE and/or BRM in vitro. Ribonucleoprotein immunoprecipitation of BRM in a *hyl1* background indicated an enrichment of main pri-miRNAs indicating that BRM acts on SE-bound pri-miRNAs before their handover to DCL1–HYL1 in vivo.

Next, the authors showed that the ATPase activity of BRM is required for inhibition of miRNA accumulation in vivo and demonstrated that BRM unwound pri-miRNAs in a way similar to dsDNA in vitro, an activity independent of the SE interaction interface yet critical for the inhibition of pri-miRNA processing in vitro. Finally, using dimethyl sulfide (DMS)-based methods, they show that BRM unwinding activity occurs *in planta* and that BRM might remodel loops and upper stems to control the productive processing of pri-miRNAs.

This work strongly suggests that BRM impedes miRNA biogenesis by acting upstream of Microprocessor in vivo.

## 6. Discussion

We have reviewed here how BRM plays a key role in many plant developmental processes and emphasized the nature and the role of direct interactors of BRM in these processes.

Several studies described here have shown that BRM is recruited to its target loci by interaction with DNA-binding proteins, such as LFY, SEP3, BP, PIFs, ERFVIIs, and TCPs transcription factors. Most of these transcription factors bind identified DNA sequences. While REF6 is not a transcription factor per se, it can bind DNA through its zinc fingers and can demethylate lysines on histones at the target genes. REF6 can also act independently of its DNA-binding through direct interaction with NF-Y factors [60,61] and it remains to know whether BRM also cooperates with REF6 at these NF-Y target genes.

The binding of REF6 precedes the recruitment of BRM [23]. This is probably also true for BRM interactions with LFY and SEP3 [12] and might be a general rule for SWI2/SNF2 ATPase recruitment in which the DNA-binding protein comes first to the target genes.

How does BRM act once it is recruited to target loci by a specific transcription factor?

Although BRM can act as a trxG factor, it can also act independently of the PRC2 complex. When BRM acts as a repressor of target loci such as in the BP-dependent regulation of *KNAT2* and *KNAT6*, the level of the H3K27me3 is unchanged but the level of H3K4me3 active mark is decreased in a BRM dependent manner, suggesting that BRM acts through PRC2-independent pathways. In some cases, BRM seems to also act through the deposition of H2A.Z which marks both transcriptionally active and inactive genes [25,62]. These chromatin modifications are likely to occur at a single locus as illustrated by the regulation of *FLC* by the BAF60/SWP73B/CHC1 subunit [63].

Most of the DNA-binding proteins that directly interact with BRM regulate genes involved in developmental processes such as vegetative development, flowering induction, inflorescence development, and flower morphogenesis (Figure 1b).

Many other DNA-binding proteins have been identified as direct interactors of the N-terminal region of BRM based on the Y2H assay [49] and it will be of interest to determine whether these proteins have the ability to co-regulate target genes with BRM *in planta*.

The second set of interactors of BRM are proteins with enzymatic activities that use BRM as substrate. These interactors (e.g., SnRK kinases, PP2Cs phosphatases and AtMMS21 SUMO ligase) and probably the RPT5a-containing 26S proteasome are involved in abiotic stresses responses. Regulation of BRM activity by phosphorylation is in agreement with the identification of several phosphorylation sites upon osmotic stress [64] and has been shown to play a direct role in drought tolerance [47]. A recent N-terminomic study also revealed potential N-terminal acetylation of BRM in seeds in relation with the Arg/N-end rule pathway [65] which also affects the stability of BRM interactors ERFVIIs [53].

The third type of BRM interactor is represented by the zinc finger RNA-binding protein SE.

This interaction demonstrates that BRM is a previously unrecognized negative regulator of miRNA accumulation for a large subset of miRNAs in Arabidopsis. This is achieved by the unwinding of pri-miRNAs by BRM in a way similar to double-strand DNA unwinding, an unexpected activity as an earlier study implies that yeast SWI/SNF complexes lack helicase activity [66]. BRM binding to pri-miRNAs must occur before SE/Microprocessor binding. The authors propose a model in which BRM inhibits miRNA production by obtaining pri-miRNAs from SE and remodeling the RNA substrates, rendering them unsuitable for processing by Microprocessor. This function might have been conserved throughout evolution as the mammalian orthologue of SE, Ars2, also participates in miRNA-dependent silencing. In humans, BRM can associate with mRNA and regulate alternative splicing [67]. In plants, the SWI3B sub-unit, physically interacts with a lncRNA-binding protein, IDN2, and contributes to lncRNA-mediated transcriptional silencing to help positioning nucleosomes on specific genomic loci [68].

## 7. Conclusions

As a chromatin remodeler, BRM is able to affect any region of the genome but can also act highly selectively due to its association with interacting factors. Further investigations should lead to a better understanding of how BRM is recruited to or released from its target genes and how its remodeling activity is regulated. Yet BRM has chromatin-independent roles, at least in RNA processing, that have been overlooked until now. Further studies will likely reveal the central position occupied by BRM and SWI/SNF complexes within regulatory networks controlling fundamental biological processes.

## Figures and Tables

**Figure 1 genes-11-00090-f001:**
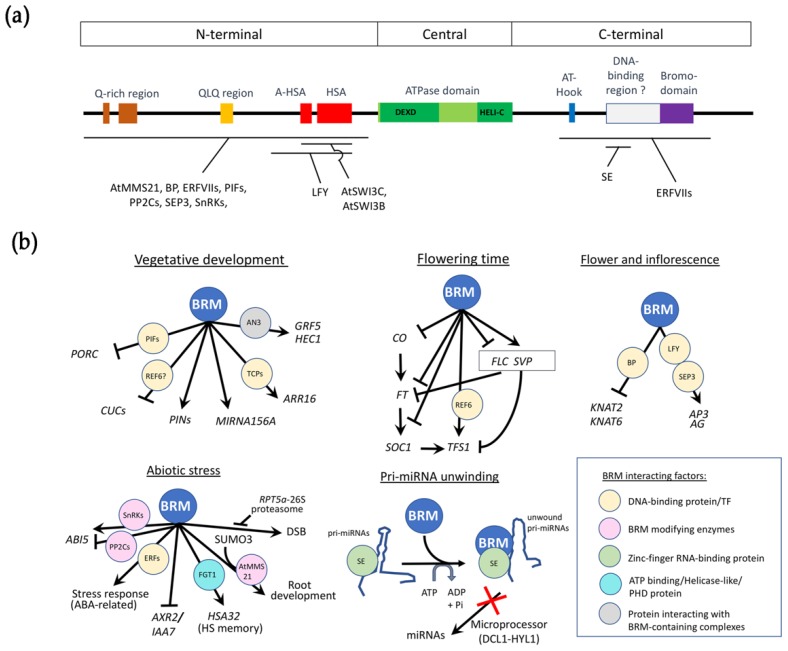
BRM domains and BRM roles in plant development. (**a**) Schematic representation of BRM domain compositions and regions that its partners interact. Partners that interact with BRM are shown below; (**b**) Overview of the main BRM roles in plant development. Gene regulations by BRM are shown by hammerhead arrows and pointed arrows, representing repression and activation, respectively. Direct interactors of BRM are presented by circles with color background according to 5 types and described in the inset. ANGUSTIFOLIA3 (AN3) may not interact directly with BRM as a direct interaction has not been reported in the literature. Core SWI/SNF subunits SWI3A, B and C that interact physically with BRM are described in [5].

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
