# Peer review of "Unwinding BRAHMA Functions in Plants"

_genes, 2020, doi:10.3390/genes11010090_

Round 1
Reviewer 1 Report
SWI/SNF (SWItch/Sucrose Non-Fermentable) is a nucleosome remodeling complex found in eukaryotes. The key component of the complex is the catalytic ATPase subunit BRAHMA (BRM). The very famous complexes have been studied for decades and known to possess multiple functions, through interaction of numerous transcriptional factors, post-translational modifiers, and newly reported microRNA processing component Serrate/Ars2. Here Thouly et al. comprehensively updated the recent progresses in studying functions and mechanisms of BRM in various biological processes. Overall speaking, the review recapitulated very well the recent findings of BRM-related activities; and it is in a good shape for publication.
This reviewer might suggest a few things for consideration:
integrate the discussion part into the preceding context to avoid redundancy. It would be nice if the authors could incorporate some recent nice work related to structure of human SWI/SNF complex into functional domain of BRM protein here (if possible). Please be consistent using the terms of SWI/SNF, SWI2/SNF2 ATPase. Is there SWI/SNF2? Some typos:
Line 47: define SPLAYED (SYD) here.
Line 52: null mutants of BRM and SYD are viable but fertile.
Line 139: single brm-101 mutant shows about 2% of embryo lethality while syd embryo lethality is close to wild-type. Brm-101 is not a null allele, correct?
SWI/SNF complexes lack helicase activity--- please change: earlier study implies that yeast SWI/SNF complexes lack helicase activity---- (the original citation is Côté, J., Quinn, J., Workman, J. L. & Peterson, C. L. Stimulation of GAL4 derivative binding to nucleosomal DNA by the yeast SWI/SNF complex. Science 265, 53–60 (1994).
Reviewer 2 Report
Thouly et al comprehensively review studies on BRAHAM functions in Arabidopsis thaliana. I enjoyed this excellent manuscript with wonderfully well-organized chapters, making me re-realized that BRAHMA is a quite attractive factor to study. This review paper will facilitate our understanding in the field of development and RNA processing in plants.
Minor comments
Line 97 - Authors stated here (caption in Figure 1) that "AN3 may not interact directly with BRM". Please include similar statement (or clarify the reason) in the paragraph describing the interaction of AN3 with SWI/SNF2 to avoid misleading that BRM is confirmed to directly interact with AN3 (line174-181).
Line 252 Please introduce more about the BRM function in root stem cell maintenance reported by Yang et al. (ref.47). They suggested that the root growth defects in brm mutant are caused by misregulation of auxin pathway. At least, in terms of root growth, BRM might have function in a crosstalk between auxin and ABA. Please discuss this.
Line 277 - 279 Please provide reference ref.45.
